

# Optimizing muscle mass and function in advanced lung cancer patients: randomized, double-blind, placebo-controlled trial protocol using High Eicosapentaenoic acid and PROtein (HEPRO) to modulate epigenetics, reduce toxicity and improve gut microbiota

Imanuely Borchardt[1,2], Carla Prado[3], Tatiane Montella[2], Gisele Fraga Moreira[2], Gisele Farias[2], Marina Xavier Reis[2], Fernanda Taveira[2], Fernanda Carneiro Dias[2], Pedro De Marchi[2], Alberto Davalos[4], Carolina Alves Costa Silva[5], Carlos Gil Moreira Ferreira[2], Andreia Melo[2] and Wilza Peres[1]

[1] Institute of Nutrition Josué de Castro of Federal University of Rio de Janeiro, Rio de Janeiro, Brazil
[2] Oncoclínicas & Co—Medica Scientia Innovation Research, São Paulo, São Paulo, Brazil
[3] Human Nutrition Research Unit, Department of Agricultural, Food & Nutritional Sciences, University of Alberta, Edmonton, Alberta, Canada
[4] Laboratory of Epigenetics of Lipid Metabolism, Madrid Institute for Advanced Studies (IMDEA), Madrid University, Madri, Spain
[5] Clinicobiome, Gustave Roussy Cancer Campus, Villejuif Cedex, France

Corresponding author
Imanuely Borchardt,
manuborchardt@yahoo.com.br

## ABSTRACT

**Background.** Lung cancer is strongly associated with malnutrition and detrimental changes in muscle mass (MM), which can lead to reduced quality of life and reduced tolerance to and efficacy of antineoplastic treatment. The loss of MM and myosteatosis (fat infiltration into muscle) have been linked to inflammation in cancer, and n-3 polyunsaturated fatty acids (PUFA) found in fish oil are known to modulate inflammatory response, lean mass, microbiota, and epigenetic mechanisms.

**Methods.** High Eicosapentaenoic acid and PROtein (HEPRO) is a randomized, double-blind, placebo-controlled clinical trial. A total of 50 patients over 20 years of age diagnosed with stage III or IV non-small cell lung cancer with an ECOG performance status of 0–2 who are eligible for systemic treatment will be included and randomized 1:1 into two treatment arms: four fish oil capsules containing 2,100 mg EPA and 924 mg DHA per day versus four placebo capsules containing 2,250 mg oleic acid per day for 4 months. All patients will be instructed to consume 1.5 g protein per kg body weight per day and receive protein supplementation if necessary. MM, myosteatosis, muscle function, handgrip strength, dietary assessment, toxicity, response to treatment, and survival will be assessed. Translational research includes membrane phospholipid composition, gut microbiota, inflammation, and miRNA. MicroRNA will be analyzed by quantitative real-time polymerase chain reaction, phospholipids, by gas chromatography, and microbiota, by 16S ribosomal RNA genetic sequencing. Statistical

analysis will be conducted using IBM SPSS Statistics V.26 and a multiple regression model will be proposed. Associations with $p < 0.05$ will be considered significant. **Conclusions**. The HEPRO study aims to evaluate a viable dietary intervention strategy to improve MM and function in patients with lung cancer.

# INTRODUCTION

Abnormalities in body composition are prevalent in lung cancer (LC) patients. They affect muscle function, contribute to the occurrence of toxicity in systemic treatment modalities, impair functionality and quality of life, and increase mortality (*Naito et al., 2017*; *Sjoblom et al., 2016*; *Cortellini et al., 2019*; *Ryan & Sullivan, 2021*; *Surov et al., 2021*). Despite advances in antineoplastic therapies based on molecular diagnosis and the use of targeted agents, LC remains the leading cause of cancer mortality worldwide, accounting for 18% of all cancer deaths (*Bray et al., 2024*). In Brazil, it is the third most common tumor type in men and the fourth most common in women (*Santos et al., 2024*). Muscle mass (MM), assessed by computed tomography (CT), is a prognostic marker that is associated with whole-body measurements and is considered the gold standard for determining body composition, as it identifies muscle depletion and attenuation even in individuals with normal or high body weight (*Mourtzakis et al., 2008*; *Baracos et al., 2010*; *Takenaka et al., 2021*; *Martin et al., 2013*).

Early and continuous nutritional intervention providing nutrients/ingredients in adequate quantity and quality is recommended to support muscle anabolism. Combinations of immunomodulatory nutrients, such as n-3 polyunsaturated fatty acids (PUFAs) and a high-protein diet, appear to yield superior nutritional outcomes (*Prado, Purcell & Laviano, 2020*; *De van der Schueren et al., 2018*). N-3 PUFAs found in fish, such as eicosapentaenoic acid (EPA) and docosahexaenoic acid (DHA), have been shown to treat MM depletion by modulating inflammatory mediators (*Schiessel et al., 2016*), epigenetic mechanisms (*Guller & Russell, 2010*), and the intestinal microbiota (*Parolini, 2019*). However, there are no effective recommendations regarding early intervention, doses, and duration in patients with cancer (*Roel et al., 2020*; *Muscaritoli et al., 2021*).

In two studies using CT to assess muscle quantity and quality in patients with cancer, n-3 PUFA supplementation favored a decrease in muscle adiposity, leading to better skeletal muscle composition (also termed "quality") a measure of myosteatosis (fat infiltration into muscle) (*Murphy et al., 2011*; *Aredes et al., 2019*). To date, most nutritional interventions for LC have focused on assessing the inflammatory response (*Tao, Zhou & Rao, 2022*; *Amiri Khosroshahi et al., 2024*). There is an urgent need for clinical trials, given the low therapeutic results in individuals with body composition abnormalities (*Sjoblom et al., 2016*; *Cortellini et al., 2019*; *Surov et al., 2021*).

The use of biomarkers represents a promising approach for the noninvasive evaluation of modifiable parameters that could improve cancer treatment outcomes. MicroRNA appears to be involved in the transcription of atrophy-related genes and increases during the proliferation of muscle components (*Guller & Russell, 2010*; *Yedigaryan et al., 2022*; *Nakasa et al., 2010*). Additionally, gut–muscle axis mechanisms affect the balance between protein synthesis and degradation, with the gut microbiota playing a role in regulating skeletal muscle mass and influencing therapeutic response (*Lahiri et al., 2019*; *Liu et al., 2021*; *Chen et al., 2023*; *Zeriouh et al., 2023*; *Clinicaltrials.gov*).

Although oncology nutrition guidelines recommend the use of n-3 PUFA to maintain or recover MM, the strength of this recommendation is weak (*Roel et al., 2020*; *Muscaritoli et al., 2021*). To date, no study has combined the use of n-3 PUFA with the protein supply recommended by international and national guidelines while evaluating skeletal muscle function and anabolic response using CT imaging in patients with LC. This study protocol aims to evaluate whether EPA-rich fish oil combined with a high-protein diet can modify skeletal muscle quantity and muscle function in patients with LC.

## MATERIALS & METHODS

This study protocol is registered at Clinicaltrials.gov NCT04965129 and approved by the ethics committee of Hospital Pró-Cardiaco (approval no. 4.486.268). It consists of a randomized, double-blind, placebo-controlled, single-center clinical trial recruiting participants over 20 years old diagnosed with non-small cell lung cancer (NSCLC) who are eligible for systemic treatment (*Riely et al., 2024*) at a private clinic in Rio de Janeiro, Brazil. Change in MM assessed by CT is the primary outcome and change in muscle function is the co-primary outcome. Secondary outcomes are MM radiodensity attenuation, concentrations of omega-3 fatty acids EPA and DHA in red blood cells, muscle strength, changes in gut microbiota composition, and changes in miRNA 1, miRNA 133a, and miRNA133b levels. The intervention will take place over 4 months, as shown in Table 1, following Standard Protocol Items: Recommendations for Interventional Trials (*Chan et al., 2017*) (See Fig. 1 for the participant timeline).

### Eligibility criteria

Inclusion and exclusion criteria are described in Table 2. All participants are required to provide written informed consent prior to any trial procedure.

### Recruitment

During routine consultations with a clinical oncologist and/or nutritionist, patients will be informed of the research objectives and invited to participate in the study following agreement from their attending physician. Upon agreeing to participate and signing the informed consent form, they will receive detailed protocol instructions.

### Randomization and blinding

Participants will be randomized in 1:1 blocks using an online free-of-charge system available at http://www.random.org, managed by a non-blinded pharmacist. All other

**Table 1  Recommendations for intervention studies (SPIRIT): schematic diagram of participant timeline.**

| | Study period | | | | | |
| --- | --- | --- | --- | --- | --- | --- |
| | Enrolment | Allocation | Post-allocation | | | Close-out |
| Timepoint | | Baseline T0 | Month 1 T1 | Month 2 T2 | Month 3 T3 | Month 4 T4 |
| Eligibility screening | x | | | | | |
| Informed consent | | x | | | | |
| AUDIT application | x | | | | | x |
| Allocation | | x | | | | |
| INTERVENTION: | | | | | | |
| 4.0 g fish oil/day | | | | | | |
| 4.0 g olive oil/day | | | | | | |
| ASSESSMENTS: | | | | | | |
| **Primary outcome:** SMI | | x | | | | x |
| Timed Up and Go test | | x | x | x | x | x |
| **Secondary outcome:** | | | | | | |
| Radiodensity | | x | | | | x |
| Concentrations of omega-3 fatty acids red blood cells | | x | | | | x |
| Muscle strength | | x | | | | x |
| 16S rRNA sequencing | | x | | | | x |
| miRNA1, miR-133a, 133b | | x | | | | x |
| **Exploratory outcomes:** | | | | | | |
| Laboratory tests | | x | | | | x |
| Anthropometry | | x | x | x | x | x |
| Food intake | | x | x | x | x | x |
| Adverse events | | x | x | x | x | x |
| Response to treatment | | | | | | x |
| Overall survival at six months | | | | | | x |
| **Adherence assessment** | | x | x | x | x | x |

**Notes.**

Abbreviations: SMI, skeletal muscle index; miR -1, Micro RNA 1; miR-133a, MicroRNA133a; miR-133b, MicroRNA133b.

parties, including patients, researchers, nurses, and technicians, will remain blinded to the treatment provided. The randomization list will be accessed only at the end of the intervention for all participants.

After randomization, as shown in Fig. 2, patients will be allocated to one of the groups, as follows: Arm A—Intervention (EPA-rich fish oil)—25 patients will receive four capsules containing one g fish oil with 525 mg EPA and 231 mg DHA, making a total of 2,100 mg of EPA and 924 mg of DHA per day, for 4 months; Arm B—Placebo (olive oil): 25 patients will receive four capsules containing one g olive oil and a total of 2,250 mg oleic acid once a day for 4 months. All the capsules have the same soft, yellowish appearance, with no identification marks. Each bottle contains 120 capsules, and intake adherence will be

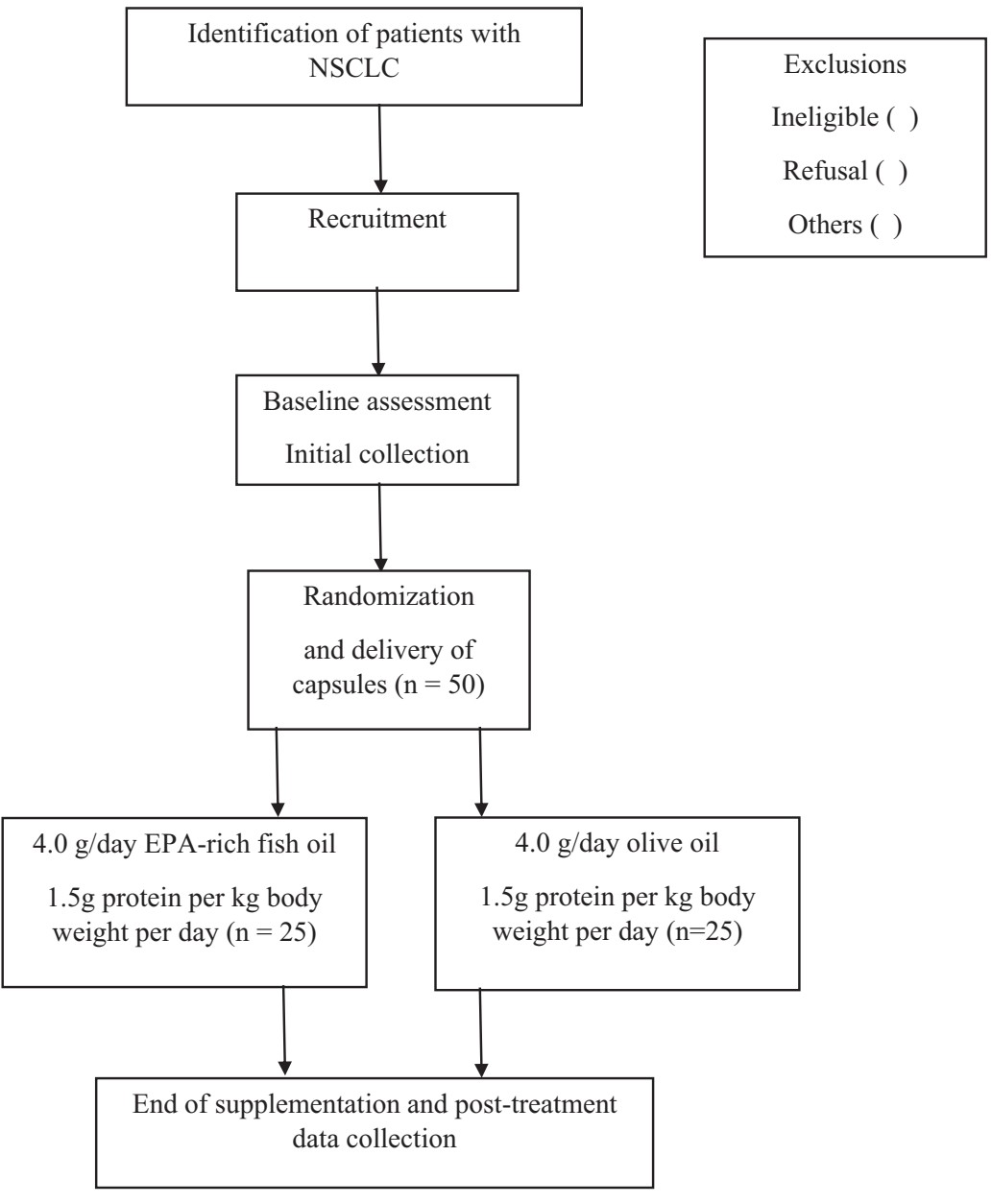

**Figure 1 Flowchart of study participants.**

assessed monthly during outpatient care visits and by telephone contact, by checking the number of capsules supplied and any leftovers.

## Intervention

During the data collection visit, the following procedures will take place: collection of blood samples; guidance for collecting stool samples; anthropometric assessment; dynamometry; timed up-and-go test; dietary assessment based on 24-hour dietary recall (24-h DR); and electrical bioimpedance. After these procedures, participants will receive one bottle

**Table 2  Eligibility criteria for the HEPROM study.**

| Inclusion criteria | Exclusion criteria |
|---|---|
| o NSCLC diagnosis<br>o Stage III and IV<br>o Age > 20 years<br>o Performance status (PS) according to ECOG 0 to 2<br>o Eligible to receive systemic treatment with immunotherapy, chemotherapy and tyrosine kinase inhibitors, combined or not with radiotherapy and/or surgery<br>o Life expectancy greater than 12 weeks | o Use of an n-3 fatty acid supplement in the last 6 months prior to the study<br>o Weight loss >10% in the last 6 months<br>o Chronic liver disease<br>o Chronic kidney disease defined as a glomerular filtration rate (GFR) of less than 60 mL/min/1.73 m$^2$, calculated according to the Chronic Kidney Disease Epidemiology Collaboration (CKD-EPI)<br>o Symptoms of previous nutritional impact that would affect the ability to follow dietary recommendations (Example: anorexia, dysphagia)<br>o Severe dietary restrictions (severe food allergy or vegetarian or vegan dietary pattern)<br>o Patients with recently diagnosed metabolic disorders (<3 months), such as uncontrolled diabetes mellitus<br>o Uncontrolled diabetes mellitus<br>o Cognitive impairment or dementia |

containing 120 capsules (EPA-rich fish oil or placebo), along with instructions on their use and storage. They will also receive an investigational product diary to record their consumption and an appointment to return monthly for evaluations and adherence checks.

Both groups will receive guidance on usual dietary care based on cancer patient guidelines, ensuring a protein intake of 1.5 g per kg of body weight per day, which is a safe dose according to recommendations for cancer patients (*Muscaritoli et al., 2021*; *Ford et al., 2024*). Nutritional counseling with specific dietary planning will be tailored individually, considering anthropometric data, and energy and protein needs to ensure daily protein consumption of 1.5 g per kg of body weight. A protein module (whey protein isolate, Nestle®, Vevey, Switzerland) will be provided to complement the protein intake. If necessary, patients indicated for oral nutritional therapy will receive a standard n-3 PUFA-free formula and/or specialized nutrients. Capsule use will be interrupted if the gastrointestinal tract cannot be used or in the event of diarrhea (three or more liquid bowel movements per day) and will be reintroduced within 7 days provided the clinical condition improves within this period. Contact will be maintained every two weeks by telephone to monitor the diet (by 24-h DR) and supplement use.

## Outcomes
### *Primary outcome*
Change in MM, assessed by CT, and change in muscle function are the two primary endpoints. Measuring muscle cross-sectional area by CT can serve as a surrogate marker for whole-body skeletal muscle mass. Meanwhile, muscle function is important in assessing functional status and can be variably affected by reduced MM. Furthermore, it is a defining

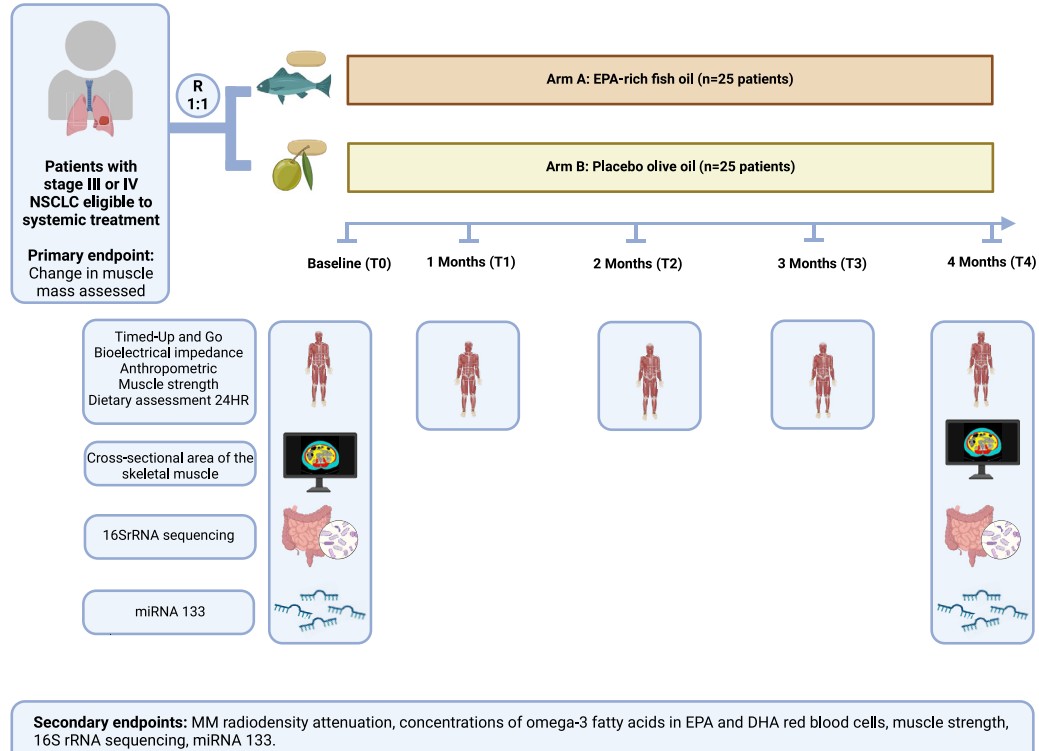

**Figure 2** **The protocol using high eicosapentaenoic acid and protein (HEPRO) study protocol.** Abbreviation: NSCLC, non-small cell lung cancer; EPA, eicosapentaenoic; DHA, docosahexaenoic; miR -1, Micro RNA 1; miR-133a, MicroRNA133a; miR-133b, MicroRNA133b; PFS, progression-free survival; OS, overall survival. This figure was created with BioRender.com.

characteristic of sarcopenia and should be investigated for its importance in individuals at nutritional risk (*Derstine et al., 2018*; *Compher et al., 2022*; *Kirk et al., 2024*).

Baseline and follow-up CT images will be used to evaluate body composition, including MM. The cross-sectional area of the skeletal muscle at the third lumbar vertebra will be measured and analyzed for muscle cross-sectional area (cm²), including the psoas, sacroiliac-lumbar, quadratus lumborum, transversus abdominis, internal oblique, external oblique, and rectus abdominis muscles (*Prado et al., 2008*). If the third lumbar vertebra is unavailable, another cross-sectional area can be used, such as T12 (*Shen et al., 2022*). The cross-sectional area will be compared between baseline (T0) and 4 months (T4).

Muscle function, as the co-primary outcome, will be assessed using the timed up-and-go test. This test provides a whole-body assessment of muscle function by having the participant rise from a standard chair, walk a distance of three m, turn around, walk back, and sit down again (*Cruz-Jentoft et al., 2019*).

The software sliceOmatic 5.0 (TomoVision, Montreal, Canada) will be used to calculate the corresponding tissue cross-sectional areas, according to the attenuation values of each tissue estimated by the Hounsfield scale (HU): from −29 to +150 for muscle; from

−190 to −30 for visceral adipose tissue; and from −150 to −50 for visceral adipose tissue (*Mourtzakis et al., 2008*).

### Secondary outcomes

*Muscle mass radiodensity attenuation.* Attenuation of MM radiodensity, also defined as myosteatosis, assesses fat infiltration into muscle using CT images. The attenuation of MM radiodensity will be classified as −29 to +29 HU for low radiodensity and +30 to +150 for high radiodensity (*Silva de Paula et al., 2018*). Images will be evaluated by the same trained observer and verified by a second observer. To minimize variations, the use of the same contrast phase will be standardized at times T0 and T4 for comparative purposes (*Morsbach et al., 2019*). Cut-off points for total, abdominal, and visceral adipose tissue will be classified according to *Doyle et al. (2013)*. All cut-off points used to identify body composition abnormalities will be revised in response to the rapid evolution in this field of study.

*Concentration of omega-3 fatty acids EPA and DHA in red blood cells.* The concentration of omega-3 fatty acids EPA and DHA in red blood cells is the most suitable method for monitoring long-term exposure in tissues, as it is the most easily accessed cell type to assess changes in EPA and DHA levels, with incorporation of EPA greater than or equal to 80% being taken to indicate good adherence to the supplementation regimen (*Aredes et al., 2019*; *Khankari et al., 2024*). The concentrations of n-3 PUFA and other fatty acids before and after supplementation will be assessed by gas chromatography at the Nutritional Biochemistry Laboratory of the Nutrition Institute at the Federal University of Rio de Janeiro. Erythrocyte samples will undergo lipid extraction, saponification, and direct alkaline methylation using an adaptation of Method 2b-11, proposed by the American Oil Chemists' Society (*American Oil Chemists Society , AOCS*). For this evaluation, 4 mg internal standard C13:0 (Sigma-Aldrich, St. Louis, MO, USA), 100 mg erythrocytes, and five mL sodium hydroxide in 0.5 M methanol will be added to the tube. The tubes will be manually agitated and placed in a shaking water bath for 15 min at 100 °C, 4 rpm. After cooling, five mL $BF_3$-$CH_3OH$ (Sigma-Aldrich, Saint Louis, MO, USA) will be added and the mixture will be placed in a shaking water bath at 100 °C for 2 min.

After cooling again, five mL hexane and three mL saturated sodium chloride will be added. The samples will be kept at room temperature for approximately 2 h for transesterification. The fatty acids will be quantified on a GC 7890A chromatograph equipped with a hydrogen flame ionization detector using Agilent OpenLab Chromatography Data System EZChrom Elite (Agilent Technologies, Inc., Santa Clara, CA, USA) and an SP 2,560 capillary column (biis-cyanopropyl polysiloxane, 100 m × 0.25 mm ID, 0.20 μm thick; Supelco®, Bellefonte, PA, USA). The fatty acid methyl esters will be identified by retention time relative to the standards (Reference Standard GLC 463, Nu-Chek Prep, Inc., Elysian, MN, USA), which are expressed as percentages in this study. The omega-3 index will be calculated by summing the percentages of EPA and DHA in the erythrocytes. Adherence to supplementation will be considered good when the index is greater than 80%.

*Muscle strength.* As an additional measurement of muscle function, upper-body muscle strength will be assessed in (kg), using a hydraulic hand dynamometer (Jamar® Hydraulic Hand Dynamometer, Sammons Preston Rolyan, Bolingbrook, IL, USA), first with the right hand and then with the left, three times each, selecting the highest measurement for comparison with the reference cutoff value. Patients will be in a seated position without arm support with their shoulders adducted and in neutral rotation. Their elbow will be flexed to 90°, with the forearm in a neutral position and the wrist varying from 0 to 30° in extension, as recommended by the American Society of Hand Therapists (*Schectman & Sindhu, 2015*; *Sousa-Santos & Amaral, 2017*). Additionally, leg muscle strength will be assessed by the chair stand test, which measures how long it takes for an individual to stand up and sit down on a chair five times without leaning on their arms (*Cruz-Jentoft et al., 2019*).

*Changes in gut microbiota composition.*

### Sample collection

Participants will receive a standard kit before the intervention (T0) and after 4 months (T4) and will be instructed to use a sterile swab to transfer a small amount of fecal sample (the equivalent of a bean) into two mL plastic microtubes containing a DNA preservation buffer solution. A sterile container will also be provided for collecting the fecal material. Samples collected in a container with DNA preservation solution can be stored at room temperature until the DNA extraction stage.

### DNA extraction

DNA will be extracted from fecal samples using the Qiagen® QIAamp® PowerFecal® DNA Kit (Qiagen, Hilden, Germany), following a standardized protocol based on the manufacturer's instructions. For extraction, each sample will be homogenized in a two mL tube with metal microbeads. This process promotes lysis of the host cells and the microbial cells, which, together with the addition of appropriate chemical reagents, ensures efficient extraction of the bacterial DNA.

The bacterial DNA extraction kit uses inhibitor removal technology for feces analysis. This technology is very effective at removing inhibitory substances commonly found in fecal material, such as polysaccharides, heme, and bile salts, which interfere with the polymerase chain reaction (PCR). After this stage, total genomic DNA is captured on a silica membrane in a column format. Following washing and elution, the isolated, high-purity DNA is obtained for genetic sequencing of the microbiota.

### Library preparation and 16SrRNA sequencing

Library preparation consists of adding identifiers to the samples. At this stage, each DNA sample will receive specific labels that allow for their simultaneous processing. At this point, the specific primer sequences containing the V3 and V4 variable regions described below will be added:

16S Amplicon PCR Forward Primer = 5′
TCGTCGGCAGCGTCAGATGTATAAGACAGCCTACGGGNGGCWGCAG

16S Amplicon PCR Reverse Primer = 5′ GTCTCGTGGGCTCGGAGATGTGTATAAGAGACAGGACTACHVGGGTAT CTAATCC

The products will be grouped and the libraries quantified by PCR using the Quant-iT and Qubit DNA quantitation assay kits (PicoGreen[®], Quant-iT[™]) and Kapa Hifi HotStart (Roche[®]) before introducing the samples into the sequencer. Intestinal microbiota composition will be determined using the 16S ribosomal RNA (16SrRNA) genetic sequencing technique, which is a next-generation sequencing amplification method used to identify and assess the abundance of bacteria. Additionally, the phylogeny and taxonomy of complex microbiome samples present in a sample will be studied.

After preparing the samples and library, bacterial DNA samples will be inserted into the sequencer (Illumina[®] MiSeq), which can provide up to 70 million paired reads, generating up to 15 GB of data. For this study, a minimum of 68,000 reads per sample was selected as the standard to better capture the microorganisms evaluated.

**Bioinformatics of intestinal microbiota data**

Data generated by 16SrRNA sequencing will be transferred, analyzed, stored, and shared on the BaseSpace platform. On this platform, sequences are compared with the Greengenes database to form amplicon sequencing variants. After the taxonomic denomination, it is possible to calculate the relative abundance of the species found and their phylogenetic classification, enabling the calculation of Simpson's index, richness, and distribution by phylum, genus, and species.

*Change in miRNA 1, miRNA 133a, and miRNA133b levels.* MicroRNAs (miRNAs) act in several pathophysiological processes, including cancer, and the evaluation of the circulating miRNA profile represents a promising approach to making non-invasive assessments of MM loss in cancer (*Yedigaryan et al., 2022*).

Total RNA will be extracted from plasma samples using the miRNeasy kit (Qiagen, Hilden, Germany), following the manufacturer's instructions. The TaqMan[®] Advanced miRNA Assay Kit (Applied Biosystems, Foster City, CA, USA) will be used to obtain cDNA. Real-time quantitative PCR will be performed using TaqMan[®] Fast Advanced Master Mix [2x] and the specific primer for the miRNA will be analyzed, in addition to the miRNA used as a normalizer, in triplicate. Quantitative reverse transcription PCR will be performed using the StepOnePlus[™] system (Thermo Fisher Scientific, Waltham, MA, USA). The relative expression of miRNA 1, miRNA 133a, and miRNA133b will be calculated using a calibration control sample. The results will be analyzed using the delta cycle threshold (ÿÿCT) method, as described by *Livak & Schmittgen (2001)*. The tests will be carried out at the Nutritional Biochemistry Laboratory of the Nutrition Institute at the Federal University of Rio de Janeiro, in Brazil, in partnership with the Lipid Metabolism Epigenetics Laboratory at the Madrid Institute of Advanced Studies, in Spain.

**Exploratory variables**

Additional demographic and clinical variables, such as tumor histology, molecular diagnosis, staging, and antineoplastic treatment will be collected. To compare

supplementation with EPA-rich fish oil *versus* the placebo, following variables will be evaluated: dietary assessment, anthropometry, electrical bioimpedance, inflammatory markers (assessed by neutrophil/lymphocyte ratio), platelet/lymphocyte ratio, toxicity, estimated response rate in 6 months, and overall survival.

Dietary assessment will be analyzed by 24-h DR on two typical days and one atypical day, and the United States Department of Agriculture's Multiple Pass Method (*Moshfegh et al., 2008*) will be used to improve the quality of data collection. During the interview, patients will be asked to report on all the types of food consumed, including drinks (except water), in the previous 24 h, how they were prepared, the quantities, and the time and place where each food or meal was consumed. The nutritional composition results for the three 24-hour periods will be de-attenuated to correct for intra-individual variability using a new statistical method, the multiple source method. This program was developed by the European Prospective Investigation into Cancer and Nutrition to estimate habitual food and nutrient intake based on two or more short-term measures, such as the 24-h DR (*Sousa-Santos & Amaral, 2017*; *Livak & Schmittgen, 2001*). The use of the protein module will be recorded in the 24-h DRs. All dietary data will be entered and analyzed for total calorie and macronutrient content.

Anthropometric variables, including body weight, height, waist, arm, and calf circumference, triceps skinfold, and arm muscle area, will be collected using standard methodology (NHANES) (*Moshfegh et al., 2008*) and the body mass index (BMI) will be estimated and stratified according to the WHO (*Haubrock et al., 2011*). Body weight and height will be measured using a Filizolâ mechanical platform scale (model 761, Hamburg, Germany) with a maximum capacity of 150 kg and a stadiometer on the same scale. An inelastic tape measure with one mm resolution will be used to measure circumferences. Skinfolds will be measured between the acromion and olecranon using a clinical calipers with one mm resolution. Calf circumference measurement will be taken with the participant in a sitting position and the cut-off points will be adjusted as proposed by *Gonzalez et al. (2021)*.

Bioelectrical impedance analysis will be carried out using a portable Biodynamics® tetrapolar device, model 450 (TBW, São Paulo, Brazil), made up of two sets of aluminum electrodes divided into four terminals. After the patient has emptied their bladder, they will rest barefoot for 5 min in the supine position with their arms relaxed and legs stretched out and apart. The electrodes will be positioned on their right foot, ankle, wrist, and hand. The distal electrode is placed on the foot at the base of the middle toe and the proximal electrode is placed just above the ankle. On the hand, the distal electrode is positioned at the base of the middle finger, just above the line of the wrist joint. Raw resistance and reactance data will be collected to calculate phase angle (*Kyle et al., 2004*; *Biodynamics, 1994*; *Janssen et al., 2000*).

The neutrophil/lymphocyte ratio will be calculated as the ratio of the absolute count of neutrophils to lymphocytes. Additionally, the platelet/lymphocyte ratio will be calculated as the ratio of the absolute count of platelets to lymphocytes (*Song et al., 2021*).

Dose-limiting toxicity will be assessed according to the Common Terminology Criteria for Adverse Events, version 5.0 (CTCAE, NCI, USA) (*Biodynamics, 1994*). Response to

treatment will be assessed according to the RECIST 1.1 criteria (*Eisenhauer et al., 2009*), and overall survival will be assessed from the start of treatment to the date of death or censoring (end of assessment without death).

### Benefits to participants

Participants will receive early nutritional assessment, intervention and follow-up. The result of this intervention can prevent nutritional losses that threaten the tolerance of cancer therapy, as well as improve the functionality and quality of life for patients with LC.

### Data management

The collected data will be recorded on standardized forms specifically designed for the study protocol and then entered into the Research Electronic Data Capture (REDCap®) data management platform (*Harris et al., 2019*).

### Statistical methods

The null hypothesis H0 is: the change in MM in the treatment arm is the same as the change in MM in the control arm. The alternative hypothesis H1 is: the change in MM in the treatment arm is greater than the change in MM in the control arm.

The sample size was calculated based on the study by *Murphy et al. (2011)*, which found that 40% of patients supplemented with n-3 PUFA gained or maintained MM. The following parameters were established: significance level of $\alpha = 5\%$ (two-sided), a power of $1 - \beta$ test $= 80\%$, and a difference in the absolute delta of the muscle cross-sectional area parameter between the groups to be relatively large (effect size > 0.8). According to the Wilcoxon–Mann–Whitney two independent groups test option in GPower 3.1, the minimum number was 21 cases per group. Assuming dropout of up to 20%, this resulted in 50 individuals for the study.

The descriptive analysis will be presented in tables, with data expressed by measures of central tendency and dispersion, appropriate for numerical variables, and by absolute and relative frequencies and percentages, in the case of categorical data. Continuous variables with normal distribution will be presented as means and standard deviations, while nonparametric variables will be described by medians and interquartile ranges, as necessary. The normality of data distribution will be assessed by the Shapiro–Wilk test and by graphical analysis of histograms.

Intention-to-treat analysis will be employed to include all randomized participants in the groups to which they were originally assigned, regardless of whether they completed the intervention according to the protocol. Per-protocol analysis will be conducted to include only those participants who completed the intervention as originally allocated. Subgroup analyses may be performed to examine the effects in specific groups of participants, such as different stages of lung cancer or varying baseline nutritional statuses.

For inferential analysis, the comparison of baseline variables between the fish oil and placebo groups will be performed using Student's $t$-test for normally distributed numerical variables the Mann–Whitney test for numerical variables that may not be normally distributed, in addition to the chi-squared ($\chi 2$) test or Fisher's exact test for categorical variables. The variation between T0 and T4 within each group will be verified using

Student's $t$-test for normally distributed numerical variables, the Wilcoxon signed-rank test for numerical variables that may not be normally distributed, and the corrected McNemar test for categorical variables. The comparison of the absolute delta (T4–T0) between the intervention and placebo groups will be performed using the Mann–Whitney test. The log rank statistic will be used to compare progression-free survival and overall survival, based on the Kaplan–Meier curves between the different strata. A linear regression analysis will be used to evaluate the relationship between the rate of muscle change and the percentage of fatty acids enriched in erythrocyte phospholipids. The significance level adopted is 5%, and the statistical analysis will be performed using SPSS Statistics V26 (IBM Inc., Armonk, NY, USA).

## DISCUSSION

This randomized, placebo-controlled clinical trial is a pioneering study to evaluate the effects of n-3 PUFA supplementation combined with a high-protein diet on MM outcomes using CT in patients with advanced NSCLC receiving systemic treatment. There is a strong call for developing high-quality randomized trials to determine the effects of nutritional interventions on clinical outcomes, emphasizing the importance of meeting nutrient needs as part of comprehensive patient care (*De van der Schueren et al., 2018*; *Arends et al., 2021*; *Lam et al., 2021*; *Ilerhunmwuwa et al., 2024*).

This study will provide early nutritional assessment and intervention, as established in guidelines for patients with cancer. The results will offer evidence on the impact of nutritional care in mitigating tolerance to cancer treatment, considering the poor therapeutic outcomes of individuals with muscle abnormalities (*Surov et al., 2021*).

Additionally, the study will investigate modifiable predictive biomarkers to understand epigenetic mechanisms such as miR-133 modulation and the role of the gut microbiota. It has been shown that conditions modifying the gut microbiota can influence therapeutic response, related toxicity, and muscle mass. To date, no study has evaluated the modulation of mi-133 by n-3 PUFA in patients with NSCLC. These results may provide new insights into the nutritional management of these patients.

## ACKNOWLEDGEMENTS

Special thanks to the participants who took the time to contribute to the research. The undergraduate students, nursing staff, pharmacists, dentists, and oncologists at Oncoclínicas & Co—Medica Scientia Innovation Research (MEDSIR) who contributed to this research.

### Funding

This study was funded by Carlos Chagas Filho Foundation for Research Support of the State of Rio de Janeiro (E-26/201.050/2021), National Council for Scientific and Technological Development (CNPq) (305224/2023-9) and Instituto Oncoclínicas. The funders had no

role in study design, data collection and analysis, decision to publish, or preparation of the manuscript.

## Grant Disclosures

The following grant information was disclosed by the authors:

Carlos Chagas Filho Foundation for Research Support of the State of Rio de Janeiro: E-26/201.050/2021.

National Council for Scientific and Technological Development (CNPq): 305224/2023-9.

Instituto Oncoclínicas.

## Competing Interests

Imanuely Borchardt has previously received honoraria from Nestle Health Science. Carla Prado has previously received honoraria and/or paid consultancy from Abbott Nutrition, Nutricia, Nestlé Health Science, Pfizer, AMRA Medical, and Novo Nordisk. Alberto Dávalos is an Academic Editor for PeerJ. The authors have no conflicts of interest to declare related to the current protocol.

## Author Contributions

- Imanuely Borchardt conceived and designed the experiments, performed the experiments, analyzed the data, prepared figures and/or tables, authored or reviewed drafts of the article, and approved the final draft.
- Carla Prado conceived and designed the experiments, analyzed the data, authored or reviewed drafts of the article, cP provides expertise and experience in selected assessments, study design, and implementation, and approved the final draft.
- Tatiane Montella conceived and designed the experiments, authored or reviewed drafts of the article, and approved the final draft.
- Gisele Fraga Moreira conceived and designed the experiments, authored or reviewed drafts of the article, and approved the final draft.
- Gisele Farias conceived and designed the experiments, authored or reviewed drafts of the article, and approved the final draft.
- Marina Xavier Reis conceived and designed the experiments, authored or reviewed drafts of the article, and approved the final draft.
- Fernanda Taveira conceived and designed the experiments, authored or reviewed drafts of the article, and approved the final draft.
- Fernanda Carneiro Dias conceived and designed the experiments, authored or reviewed drafts of the article, and approved the final draft.
- Pedro De Marchi conceived and designed the experiments, authored or reviewed drafts of the article, and approved the final draft.
- Alberto Davalos conceived and designed the experiments, authored or reviewed drafts of the article, and approved the final draft.
- Carolina Alves Costa Silva analyzed the data, prepared figures and/or tables, authored or reviewed drafts of the article, and approved the final draft.
- Carlos Gil Moreira Ferreira conceived and designed the experiments, authored or reviewed drafts of the article, and approved the final draft.

- Andreia Melo conceived and designed the experiments, analyzed the data, authored or reviewed drafts of the article, and approved the final draft.
- Wilza Peres conceived and designed the experiments, performed the experiments, analyzed the data, authored or reviewed drafts of the article, and approved the final draft.

### Human Ethics

The following information was supplied relating to ethical approvals (i.e., approving body and any reference numbers):

Research Ethics Committee of Hospital Pró-Cardiaco, No. 4.486.268.

### Data Availability

This is a registered report.

### Clinical Trial Registration

The following information was supplied regarding Clinical Trial registration:

Clinicaltrials.gov

nct04965129.

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
