# Peer review of "Optimizing muscle mass and function in advanced lung cancer patients: randomized, double-blind, placebo-controlled trial protocol using High Eicosapentaenoic acid and PROtein (HEPRO) to modulate epigenetics, reduce toxicity and improve gut microbiota"

_PeerJ, doi:10.7717/peerj.19506_

## Round 0.1 · original submission · Major Revisions

Dear Authors

Even though one of the referees has recommended rejection, this in part seems to be because they didn't realize this is a Registered Report article. I would like to give you an opportunity to revise the paper based on the comments of all referees.
I am looking forward to receive you revised paper

Reviewer 1 ·

Basic reporting

Clarity and Rigor: While the manuscript is well-written in professional English, the text lacks engagement and depth in certain sections, particularly the discussion, which tends to repeat established knowledge rather than provide new insights or highlight the unique aspects of the study.
• Context and Background: The background is thorough, providing relevant references and contextualizing the study within the broader scope of nutrition interventions for non-small-cell lung cancer (NSCLC). However, it could be more concise to avoid redundancy.
• Figures and Tables: The figures (e.g., Figures 1 and 2) and tables (e.g., Table 1 and Table 2) are visually clear and informative, but they only outline the trial protocol. There are no trial results presented, which significantly limits the value of the study in its current form.

Experimental design

Originality and Scope: The study design—a randomized, double-blind, placebo-controlled trial assessing the effects of high eicosapentaenoic acid (EPA) and protein supplementation—is conceptually sound and addresses a relevant topic in cancer care. However, the protocol lacks novelty as similar interventions have been explored in the literature.
• Sample Size and Rationale: The justification for a sample size of 50 patients (25 per arm) is based on prior studies, but it remains small for assessing complex outcomes like muscle mass and gut microbiota composition. This raises concerns about the statistical power of the study to detect meaningful differences.
• Incomplete Trial: My main concern is that this manuscript only describes the trial design without reporting any results. As such, it cannot contribute substantively to the scientific literature at this stage. Submissions to PeerJ should include completed studies with sufficient data to support conclusions.

Validity of the findings

Lack of Results: The manuscript is essentially a study protocol, with no findings reported. While protocols are valuable for planning and transparency, this submission does not present any data or analyses to validate the study’s objectives or outcomes.
• Relevance of the Outcomes: The planned outcomes, such as changes in muscle mass, gut microbiota, and miRNA expression, are clinically relevant. However, without results, it is impossible to assess whether the study achieves its objectives or offers any practical implications for NSCLC management.
• Discussion Limitations: The discussion section reads more like a general review of existing literature rather than an interpretation of findings from the current study. This misalignment weakens the manuscript’s relevance and impact.

Additional comments

1. Trial Status: If this is a protocol submission, it should clearly indicate that no results are included. The authors should consider resubmitting after the trial is completed and data are analyzed.
2. Ethical Approval: Ethical approval and registration are appropriately detailed, which supports the integrity of the study.
3. Suggestions for Improvement: The authors could include preliminary or interim findings to enhance the manuscript’s value and relevance. If no results are available yet, the manuscript should be reframed as a protocol paper.

·

Basic reporting

The study is extremely important for nutrition in oncology. Congratulations to the authors for the idea and design of the study.

The manuscript is well written, clear and objective. Relevant information is described in context, with updated and reliable references.

Experimental design

I would like to suggest that the authors consider assessing muscle strength as the primary outcome, since strength responds more quickly to nutritional intervention than the amount of muscle mass. Even if there is an assessment of muscle function as the primary outcome.

It is also interesting to review the assessment time of the 24-hour recall, since, with the side effects of the treatment, there may be significant changes in food intake every 2 weeks. Another reason would be to check protein intake weekly, with the aim of ensuring 1.5g/kg

Validity of the findings

The study should consider the treatment performed, since the type of toxicity and when toxicity occurs differs between some drugs.

Describe patients who require caloric nutritional intervention, as some patients may require hypercaloric oral nutritional supplements at the beginning or during treatment.

Reviewer 3 ·

Basic reporting

o The organization of abstract needs improvement
 Line 34-38: “study aims to evaluate the …. with non-small cell lung cancer (NSCLC).” should be in “Methods” section of abstract.
 Line 45-46: please specify the primary endpoint, and secondary endpoint of the clinical trial.
 Line 48-52 may be redundant in an abstract and could be moved to main text
 Conclusion may not be applicable in a prospective trial
o Some sections have section number while other don’t. Need to be consistent
o The organization of “Materials & Methods” section needs to be improved
 At the beginning of this section, the authors should mention the overall study design including what treatment will be used, what is the key characteristics of patient population, treatment information (control and treatment intervention) and duration, study endpoints, then elaborate patient population (eligibility criteria)
 “Benefits to participants” and “Data management” may not be included in “Outcomes” section as this section describes the primary, secondary and exploratory endpoints
 Sample size and statistical methods should be moved to a separate section named “Statistical analysis”.
 There are descriptions within outcomes section about how primary, secondary and exploratory outcomes will be evaluated and how these data will be collected. These descriptions may be moved to “Intervention” section that is for evaluations/assessments in the study.
 “Intervention” section can be renamed to “Assessments” or “Evaluations”
o The “Discussions and Conclusions” section may be renamed to “Discussions” because the trial has not been started and no results have been reported, so no conclusions
o In general, the language can be improved to ensure that readers can clearly understand your text. And the organization of the paper needs to be polished. Some examples where the language could be improved include
 Line 34-38: “… using computed tomography….”, suggest to say: the primary endpoints are … , secondary endpoints are…
 Line 45:typo, what is “ood consumption?”
 Line 46-47: Grammar issue: “Translational research includes membrane phospholipid composition, gut microbiota, inflammation and miRNA 133 will be assessed”.
 Check grammar in Line 59-60, Line 81-83, line 85-87, line 96 “the strength … is weak or non-existent”
 line 91 “Addition,” should be “Additionally”
 Line 101-104: “exploratory objectives will be used to evaluate…”, grammar issue, it is not the objectives that will be used to evaluate.
 Check grammar issue in Line 108 – 111, Line 347-349,
 line 363-364: “the comparison… will be analyzed”, should be “comparison will be conducted”

Experimental design

o Definition of primary endpoints (outcomes) and secondary/exploratory endpoints in section 2.5 needs clarifications
 The primary, secondary and exploratory endpoints should be stated in English language instead of only listing them as section titles. For example, there should be a sentence at the beginning of section 2.5.1, stating that “The change in muscle mass assessed by CT and the muscle function are two primary endpoints”.
 The authors should clarify the relationships between body composition, cross-sectional area and muscle mass/muscle function.
 The definitions for the endpoints need clarifications. For primary endpoint “change in muscle mass”, how was it calculated? Is it defined as the change from baseline in muscle mass at month 4? And how to define baseline? And what is the relationship between cross-sectional area and muscle mass and muscle function?
How to calculate primary endpoint “muscle function” based on Timed-Up and Go (TUG) and gait speed?
And for the secondary outcome Muscle mass radiodensity attenuation, the clear definition and how it will be evaluated needs to be added. Same for other secondary/exploratory outcomes.

And muscle function is called “co-primary endpoint”. From statistics perspective, co-primary endpoint is a concept proposed for multiple testing, both co-primary endpoints are tested to declare statistical significance. If sample size calculation of this study is based solely on one of the primary endpoints, I would suggest renaming both as “primary endpoint”, instead of co-primary endpoint. And it should be noted in the manuscript that the sample size will be calculated based on only one of primary endpoints.

o Sample size calculation should be clarified:
 From statistics perspective, to calculate sample size needed to achieve a desired power while controlling the type I error rate, we need to define the null hypothesis and alternative hypothesis based on primary endpoint, in this paper it is the change in muscle mass assessed by CT (if this is what the authors planned). The null hypothesis H0 is: the mean change in muscle mass in treatment arm is the same control arm. The alternative hypothesis H1 is: the mean change in muscle mass in treatment arm is greater than control arm.
 But in this paper, no null and alternative hypothesis are defined. And it is not clear how “40% of patients supplemented with n-3 PUFA gained or maintained muscle mass.” is used in making the assumptions, considering the primary endpoint is a continuous variable. And “effect size > 0.8” needs clarifications as well.
o Statistical methods:
 Line 364-365: “the comparison of baseline variables between the groups (cases and controls) will be analyzed using Student’s t-test for independent samples or Mann-Whitney for numerical variables”. This is not accurate. Student t test is for normally distributed independent samples, Mann-Whitney test is for two independent samples that may not necessarily follow normal distribution.
 Line 367-368: it’s unclear what “the comparison of evolution (deltas)” is.
 Line 366-367: “the variation from start to finish in the numerical variables within each group will be assessed using the Wilcoxon signed rank test”, it is unclear what variation from start to finish is and how it can be evaluated by signed rank test.
 Line 356: does “central tendency and dispersion” mean “mean and standard deviation”?
 Line 364: use treatment and control instead of “case and control”
 Line 373 “The criterion for determining significance will be 5%.”. should be statistical significance level will be 5%.

Validity of the findings

o This is a prospective clinical trial and no findings have been reported. Please refer to comments in experimental design section.

---

## Round 0.2 · Minor Revisions

Dear Authors

Kindly make the minor changes suggested by the referee and resubmit.

Reviewer 3 ·

Basic reporting

Please see additional comments

Experimental design

Please see additional comments

Validity of the findings

Please see additional comments

Additional comments

The authors have responded and addressed most of the comments in the first round of review, except for one minor comment: in Line 382 “the Mann-Whitney test for nonparametric numerical variables,” should be “the Mann-Whitney test for numerical variables that may not be normally distributed”, line 385 “the Wilcoxon signed-rank test for nonparametric numerical variables” should be "the Wilcoxon signed-rank test for numerical variables that may not be normally distributed".

---

## Round 0.3 · accepted · Accept

Dear Authors
I have gone through the referee comments and determined that the paper is now ready to be accepted. Wishing you the best as you conduct this research and looking forward to reading the results of your trial in future.

Reviewer 3 ·

Basic reporting

The authors have addressed all comments.

Experimental design

The authors have addressed all comments.

Validity of the findings

The authors have addressed all comments.